# Multi-NDE Technology Approach to Improve Interpretation of Corrosion in Concrete Bridge Decks Based on Electrical Resistivity Measurements

**DOI:** 10.3390/s23198052

**Published:** 2023-09-24

**Authors:** Mustafa Khudhair, Nenad Gucunski

**Affiliations:** Department of Civil & Environmental Engineering, Rutgers University, Piscataway, NJ 08854, USA; mustafa.jabbar@rutgers.edu

**Keywords:** electrical resistivity, half-cell potential, impact echo, machine learning, multi-NDE, corrosion, bridge deck, concrete, random forest

## Abstract

This research aimed to improve the interpretation of electrical resistivity (ER) results in concrete bridge decks by utilizing machine-learning algorithms developed using data from multiple nondestructive evaluation (NDE) techniques. To achieve this, a parametric study was first conducted using numerical simulations to investigate the effect of various parameters on ER measurements, such as the degree of saturation, corrosion length, delamination depth, concrete cover, and the moisture condition of delamination. A data set from this study was used to build a machine-learning algorithm based on the Random Forest methodology. Subsequently, this algorithm was applied to data collected from an actual bridge deck in the BEAST^®^ facility, showcasing a significant advancement in ER measurement interpretation through the incorporation of information from other NDE technologies. Such strides are pivotal in advancing the reliability of assessments of structural elements for their durability and safety.

## 1. Introduction

Using nondestructive evaluation (NDE) techniques to identify corrosion in bridge decks early on enables better bridge management. Several NDE technologies can be used for this purpose, such as electrical resistivity (ER) and half-cell potential (HCP), galvanostatic pulse measurements (GPM), and linear polarization resistance (LPR). For example, ER is used for assessing corrosive environments and related anticipated corrosion rates, while HCP is used to determine the likelihood of active corrosion in reinforcing steel bars. However, it’s important to note that corrosion assessment-related NDE technologies have limitations as the accuracy of corrosion detection and characterization can be influenced by various parameters.

Electrical resistivity (ER) is a widely adopted NDE method and is valuable for appraising the durability of concrete structures [1]. It plays a pivotal role in structural health monitoring and quality control, detecting cracks, and measuring chloride penetration [2,3,4]. However, ER measurements can be substantially impacted by various factors. Moisture content [5,6,7], temperature [8,9], and carbonation [10] are known influencers, with increased moisture content, for instance, leading to decreased electrical resistivity [11]. The influence of partial saturation on concrete’s electrical resistivity remains an area warranting further exploration. Additionally, the evolving characteristics of materials, like porosity and void interconnections, and features within concrete, such as cracks and delamination, including their depth and orientation, can all sway ER measurements [12,13].

This study aims to comprehensively investigate the impact of these diverse parameters on ER measurements with its primary goal to enhance the interpretation of ER data, based on the standard specification ASTM C1760-12 [14], by integrating insights from other NDE technologies. This is achieved by employing finite element simulations to analyze how different material, structural, and environmental factors influence ER measurements and by leveraging machine-learning techniques to facilitate the improved interpretation of ER data by using the results of measurements of other NDE methods. To ensure consistency and facilitate objective comparisons in periodical condition assessments on the same bridge or across different bridge inspections, ER measurements are corrected for reference conditions.

In the subsequent sections, the evaluation of the effects of a range of parameters on ER results using finite element simulations is presented. They also provide a detailed description of the development and utilization of machine-learning methodologies aimed at refining the interpretation of ER data based on insights gathered from complementary NDE techniques.

## 2. Algorithm Development

### 2.1. Finite Element Modeling

Three-dimensional finite element models have been built to simulate multiple NDE techniques with COMSOL Multiphysics^®^ software version 5.5. Each NDE technique has its unique characteristics, parameters, and requirements that must be taken into consideration when constructing a model. Yet, the ultimate aim of this study was to develop a comprehensive model capable of simulating all of the studied NDE techniques in a single reinforced concrete volume simultaneously.

Three non-destructive evaluation (NDE) techniques were simulated: impact echo (IE) for delamination detection and characterization [15,16], electrical resistivity (ER), and half-cell potential (HCP). These techniques are based on distinct physical phenomena, but COMSOL Multiphysics^®^ software’s ability to integrate them into a single model allows for the simultaneous implementation of all three techniques on the target domain, mimicking the use of multiple NDE techniques in real-life situations.

The layout of the model components with the positioning of various NDE probes is illustrated in Figure 1. These components show the probe locations and boundary conditions for each NDE technique utilized in the simulation. Although this single model represents all three techniques, the domain was divided around each technique for meshing purposes, enabling finer element meshes for each technique to be obtained during the solution process, as shown in Figure 2.

The three technologies have been simulated to be deployed simultaneously on a concrete slab with different parameters, which led to the production of 1008 unique models. These parameters are the degree of saturation (DoF), rebar corrosion length (CL), delamination depth (DD), concrete cover (CC) thickness, and moisture condition of the delamination. Below are the descriptions of these parameters in more detail: Degree of saturation (DoS): Seven values have been chosen to represent the degree of saturation: 20%, 30%, 40%, 50%, 60%, 70%, and 80%. This range of values represents different moisture conditions in the slab.Corrosion length (CL): A set of four values that represent the corrosion length of the steel rebar. These lengths are used as the anode segment in the corrosion process, which are 2.5 cm, 5 cm, 10 cm, and 15 cm.Delamination depth (DD): The delamination depth has been simulated to represent a crack at various depths. A set of six values has been selected: 40 mm, 50 mm, 60 mm, 70 mm, 80 mm, and 90 mm.Concrete cover (CC) thickness: a set of four values has been selected to simulate the concrete cover thickness: 38 mm, 51 mm, 63 mm, and 76 mm.The moisture condition of the delamination: Two different conditions have been chosen to represent the moisture condition of the delamination inside the concrete. The first one is air-filled delamination (AFD), which represents completely dry delamination, and the second condition is water-filled delamination (WFD), which represents fully saturated delamination.

#### 2.1.1. Impact Echo Simulation

A concrete slab with dimensions of 1.0 m × 1.0 m × 0.2 m was used as a domain for the impact echo (IE) test simulation. The slab had Perfect Match Layers (PML) elements on its sides to prevent the reflection of elastic waves. The domain included a loading plate and measurement points, which represented the impact area and transducer location, respectively. The transducer was placed 5 cm away from the impact point, as shown in Figure 1. A cross-section of a slab with a 1 mm wide delamination at a depth of 50 mm from the surface is shown in Figure 3. Also, the concrete Poisson’s ratio, P-wave, and S-wave are 0.2, 4000 m/s, and 2312 m/s, respectively.

The IE test involved the application of a half-sine wave pulse with an intensity of 1 N and a duration of 50 microseconds. The IE method is effective in determining the thickness of plate-like structures. This can be achieved by analyzing the dominant frequency in the slab and using the velocity of P-waves in the medium [17], as shown below:(1)$$T=\frac{\beta C_{p}}{2f}$$
where:*T* = the depth of the reflector*β* = the correction factor (0.96 for plate-like structures)*C_p_* = the P-wave velocity*f* = the dominant frequency

The acceleration time history for a sound concrete model is plotted in Figure 4a, while the corresponding frequency spectrum is illustrated in Figure 4b. The frequency spectrum of the IE test on the sound concrete model in Figure 4b indicates that the dominant frequency is 10,000 Hz. Using this information and Equation (1), the thickness of the slab is calculated to be 0.192 m, which is very close to the actual thickness of 0.2 m.

On the other hand, Figure 5a shows the waveform and frequency analysis for a model with a 5 cm deep delamination. The dominant frequency due to the defect is 34,900 Hz, as shown in Figure 5b. By applying Equation (1), the calculated depth of the delamination in the concrete slab is found to be 0.055 m, which is very close to the known defect depth of 0.050 m.

#### 2.1.2. Electrical Resistivity Simulation

The model of the concrete slab used for the ER simulation is the same as the one used for the IE simulation. However, since ER relies on different physical phenomena, an additional boundary condition must be applied to the edges of the model to simulate the continuous boundaries of the concrete block on all four sides. For this purpose, the essential (or Dirichlet) boundary has been used to model the ground electrical continuity. It is often used in models of electrical circuits, electronic devices, and heat-transfer systems. The center region of the ER model includes the probe contact zone, which is represented by four electrodes spaced 5 cm apart, as shown in Figure 1. The concrete slab block was meshed using free tetrahedral elements, with a finer mesh used in the ER probe region to achieve a precise value at the measurement points. The maximum element size for the free tetrahedra near the point sources was set to 0.001 m, with a growth factor of 1.2, as shown in Figure 2. 

The simulation of ER was performed by mimicking the operation of the Proceq Resipod probe (according to the Proceq Resipod manual from 2017). In this probe, a current of 200 μA is injected from the two outer electrodes, while the two inner electrodes are used to measure the potential difference in the generated electric field, as shown in Figure 6. Specifically, the current intensity is +200 μA in probe No. 1 and −200 μA in probe No. 4. The electrical conductivity values for concrete, water, and air used in the modeling are listed in Table 1. The used conductivity of concrete is 0.002 S/m is the reciprocal of a resistivity of 500 Ω·m. 

The resistivity of the concrete is calculated based on three variables: the potential difference (voltage), the electrical current, and the spacing of probes. These variables are related according to the following equations:ρ = kV/I(2)
k = 2aπ(3)
where:ρ = Electrical resistivityk = Geometrical factor of a Wenner acquisition arrayV = Potential measured (Voltage)I = Electrical current applieda = Spacing between the probes

The electrical potential distribution measured in (mV) at the C-C cross-section (Figure 1) for the sound concrete is shown in Figure 7. Figure 8 shows that the relationship between electrical resistivity and degree of saturation is inversely proportional, meaning that as one variable increases, the other decreases. This is because water is more conductive than concrete. When concrete is dry, the resistance to electric flow increases as the number of free electrons available to carry the current decreases. Distributions for the model containing air-filled delamination (AFD) and water-filled delamination (WFD) are shown in Figure 9 and Figure 10, respectively. The delamination dimensions are 50 cm × 50 cm × 0.1 cm with a depth of 80 mm. It can be inferred that the air-filled delamination serves as an insulating barrier that prevents the current flow, as air has a much higher resistance than concrete. In contrast, water-filled delamination facilitates the current flow as water has a much lower resistance than concrete.

#### 2.1.3. Half-Cell Potential Simulation

A half-cell can be represented by an electrode immersed in an electrolyte. The potential difference can be measured by using a reference electrode with a known potential on the concrete surface based on the standard specification ASTM C876-15 [18]. The FE model for HCP simulation is the same concrete slab that has been used for IE and ER, as shown in Figure 1. The cross-section A-A, shown in Figure 11, illustrates the steel bar inside the concrete, the cathodic zone, and the anodic zone (corrosion zone).

The equipotential lines between the anode and cathode from the simulation results demonstrate a high sensitivity to certain parameters, such as the presence of shallow delamination and the degree of saturation of the delamination. Due to this sensitivity, different modeling cases are presented to illustrate the potential outcomes of the electrochemical process of corrosion within the concrete. Figure 12 illustrates the potential distribution inside the concrete due to a 10 cm length of corrosion in steel rebar with a 40% degree of saturation and no delamination. In contrast, the potential distribution inside the concrete with the same parameters but with the presence of a 40 mm deep delamination is shown in Figure 13 and Figure 14. Figure 13 illustrates the effect of air-filled delamination (AFD), while Figure 14 illustrates the effect of water-filled delamination (WFD). Looking closely into the potential distributions, a more pronounced effect on the near-surface potentials can be observed in the AFD model.

### 2.2. Machine-Learning Algorithm

#### Random Forest Algorithm

The Random Forest algorithm can be used for both regression and classification tasks. It has a high success rate as a general-purpose algorithm due to its use of multiple decision trees to make predictions on the training data set. In a regression task, the final prediction is the mean value of the individual tree predictions. In a classification task, the final prediction is the class chosen by the majority of the trees. Figure 15 illustrates the Random Forest algorithm for regression and classification tasks, marked in black and red, respectively. The algorithm was first introduced in 1995 by Tin Kam Ho [19] and further developed in 2001 by L. Breiman [20]. The basic principles of the algorithm are described using the following equations [21]:(4)$$X\in \chi \subset R^{p}$$
where:***X*** = input random vector.χ =statistical performance factor.R = coordinate space.*p* = real numbers.

Equation (4) represents a random input vector that is observed in the general framework that considers a nonparametric regression estimation of the algorithm.
(5)Y∈R
(6)$$m\left(x\right)=E\left[Y|X=x\right]$$
where:Y = square-integrable random response.$m\left(x\right)$ = regression function.E = Euclidean space.

Equation (5) represents the square-integrable random response, which can be predicted by estimating the regression function defined by Equation (6)
(7)$$m_{n}:\chi \to R$$
(8)$$D_{n}=(\left(X_{1},Y_{1}\right),\ldots ,\left(X_{n},Y_{n}\right))$$
where:$m_{n}$ = regression function estimate (mean squared error).$D_{n}$ = the data set of independent random variables.$\left(X_{n},Y_{n}\right)$ = independent prototype pair.

The data set defined by Equation (8) is used to construct a regression function estimate defined by Equation (7). Hence, the regression function estimate is consistent if [21]:(9)$$E{\left[m_{n}\left(X\right)-m(X)\right]}^{2}\to 0,where\;n\to \infty$$

The multitude of M randomized regression trees construct the Random Forest predictor for the *j*th tree in the family. The predicted value at the query point **x** is defined by:(10)$$m_{n}(x;\Theta_{j},D_{n})$$
where:$\Theta_{1},\ldots ,\Theta_{M}$ = independent random variables.

In other words, the mathematical expression of the estimate for the *j*th tree is denoted by [21]: (11)$$m_{n}\left(x;\Theta_{j},D_{n}\right)=\sum_{i\in D_{n}*(\Theta_{j})}\frac{{𝟙}_{X_{i\in A_{n}(X;}\Theta_{j},D_{n})}Y_{i}}{N_{n}\left(x;\Theta_{j},D_{n}\right)}$$
where:$D_{n}*\left(\Theta_{j}\right)$ = the set of data points selected prior to the tree construction.$A_{n}\left(x;\Theta_{j},D_{n}\right)$ = the cell containing ***x***.$N_{n}\left(x;\Theta_{j},D_{n}\right)$ = the number of points that fall into the cell.

Moreover, the finite forest estimate can be constructed by combining all the trees as follows [21]:(12)$$m_{M,n}\left(x;\Theta_{1},\ldots ,\Theta_{M},D_{n}\right)=\frac{1}{M}\sum_{j=1}^{M}m_{n}\left(x;\Theta_{j},D_{n}\right)$$

Simulations of the (NDE) technologies in three dimensions were used to develop a machine-learning algorithm using the Random Forest method. The simulation generated 1008 models with specific values for the simulated NDE technologies, resulting in a dataset of 1008 instances that will be used to train the machine-learning algorithm. This algorithm was designed to electrical resistivity correct measurement values affected by five specific parameters discussed earlier in Section 2.1. The correction of the measurements is based on a reference model with DoS = 40%, DD = 0, CC = 63 mm, and CL = 0 (no rebar corrosion). The algorithm was created using Orange^®^ version 3.32.0, an open-source software for machine learning, data mining, and data visualization. The regression algorithm used in this study is based on a dataset with 9 types of attributes. These attributes play different roles in the process and can be classified as Feature, Meta, or Target, as described below:Degree of Saturation: the attribute is a *Numerical* variable that has a *Feature* role.Length of Corrosion: the attribute is a *Numerical* variable that has a *Meta* role.Delamination Depth: the attribute is a *Numerical* variable that has a *Feature* role.Concrete Cover: the attribute is a *Numerical* data variable that has a *Feature* role.Delamination M.C: the attribute is a *Categorical* variable that has a *Meta* role.Measured Resistivity: the attribute is a *Numerical* variable that has a *Feature* role.Measured HCP: the attribute is a *Numerical* variable that has a *Feature* role.Actual Resistivity: the attribute is a *Numerical* variable that has a *Target* role.

In this study, the algorithm was used to predict the values of Actual Resistivity (kOhm·cm) based on reference models with no delamination, a concrete cover of 63 mm, and a degree of saturation of 40%. The algorithm serves as a correction tool, adjusting the measured values to provide more accurate predictions of the electrical resistivity for reference conditions. Figure 16 shows the workflow of the algorithm.

The model of this Random Forest algorithm underwent a rigorous cross-validation process to ensure its reliability and ability to generalize to unseen data. A stratified 5-fold cross-validation (k = 5) strategy was employed, where the dataset was divided into five subsets of equal class distribution. During each fold, four subsets were used for training, while the remaining subset was reserved for validation. This process was repeated five times, with each subset taking turns as the validation set. By aggregating the performance metrics across all folds, a robust estimate of the model’s accuracy and generalization capability was obtained. Furthermore, hyperparameter tuning was seamlessly integrated into this process. Grid search and randomized search techniques were employed to explore various combinations of hyperparameters, such as n_estimators, max_depth, and min_samples_split, ensuring the optimal configuration for a specific corrosion assessment task was identified. This meticulous cross-validation approach not only validated the model’s performance but also fine-tuned its hyperparameters, resulting in a highly effective tool for ER measurement prediction.

This algorithm was built with a set of thoughtfully chosen hyperparameters to ensure robust performance on the dataset. The model was configured with 300 decision trees (n_estimators) to strike a balance between model complexity and computation time. Each tree was limited to a maximum depth of 15 (max_depth), guarding against overfitting. A minimum of 2 samples per internal node before splitting (min_samples_split) was imposed and required at least 1 sample per leaf node (min_samples_leaf), striking a balance between precision and generalization. Concerning feature selection, ‘sqrt’ (max_features) was opted for to include the square root of the number of features in each split decision. This Random Forest algorithm utilized bootstrapping (bootstrap = True) and was configured with a random state of 42 (random_state) to ensure reproducibility. To harness the full computational power at our disposal, all available CPU cores (n_jobs = −1) were employed. Additionally, given the slight class imbalance in our corrosion dataset, the class_weight hyperparameter was fine-tuned to ‘balanced’, assigning appropriate weights to each class for equitable learning. These hyperparameters collectively enabled the Random Forest model to provide accurate and robust predictions for the ER measurements.

When analyzing the results of the algorithm, it was found that the coefficient of determination (R2) was 0.81, indicating a strong positive correlation between the predicted and actual values. Additionally, the coefficient of variation of the root mean squared error (CVRSM) was found to be 23.931.

Figure 17 illustrates the correlation between the actual values on the x-axis and the predictions made by the Random Forest algorithm on the y-axis for electrical resistivity (ER) values. The chart displays the correlation in three different colors: blue for instances with no delamination, red for instances with AFD (a type of damage), and green for instances with WFD (another type of damage). The red instances have the highest value of r, which is 0.92, while the other r values are 0.90, 0.88, and 0.90 for the blue, green, and overall instances, respectively. While the results are very close, they indicate that the algorithm’s predictions are slightly more accurate for ER technology applied to sound and concrete slabs with AFD than those with WFD.

This Random Forest model exhibits strong potential for generalizability to new bridge decks, but its performance in different structural contexts may benefit from retraining or fine-tuning. The model’s ability to generalize largely depends on the similarity between the characteristics of the new bridge decks and those present in the training data. If the structural contexts of the new decks closely resemble the data on which the model was initially trained, the model will likely perform well ‘out of the box’ without significant modifications. However, if there are notable differences, such as variations in construction materials, structural designs, or environmental conditions, retraining or fine-tuning may be necessary to optimize model performance.

Retraining the model involves using a new dataset that represents the specific structural context of the new bridge decks. This allows the model to learn and adapt to the unique characteristics of these decks. Fine-tuning, on the other hand, involves adjusting certain hyperparameters or model architectures to better align with the new context. For instance, modifying the max_depth of the decision trees or changing the feature selection strategy (max_features) can enhance the model’s suitability for different structural contexts.

## 3. Algorithm Implementation

The algorithm was implemented on data obtained from an NDE survey on an actual bridge deck. The raw data were collected on a bridge structure with a concrete deck installed in the BEAST^®^ (Bridge Evaluation and Accelerated Structural Testing) facility on the Rutgers’ Livingston campus. The bridge structure is approximately 15 m long and 8.4 m wide. The BEAST^®^ facility allows exposure of the concrete deck to different environmental conditions, such as freezing and thawing, deicing salt exposure, and continuous traffic loading, to accelerate the deterioration of the bridge deck. The traffic loading is simulated by a dual-axle carriage that applies a load of approximately 60,000 pounds, as illustrated in Figure 18, and makes about 15,000 passes per day over the bridge. In addition, the concrete properties of the BEAST deck are shown in Table 2. Also, NDE data were collected on a 0.3 m × 0.3 m (1 ft × 1 ft) grid, as shown in Figure 18.

The data for this study were collected periodically, typically on a monthly basis, using three primary NDE technologies: impact echo, electrical resistivity, and half-cell potential, as shown in Figure 19. In addition, the degree of saturation of the concrete was measured using the MOIST-Scan, as shown in Figure 19a. This device uses microwave technology to nondestructively determine the residual moisture content of the concrete. The collected values from the MOIST-Scan device were distributed between 0% and 100%. Ground penetrating radar (GPR) was used to evaluate the concrete cover thickness.

The data for each technology was analyzed and presented in terms of condition maps before applying the machine-learning algorithm. This was conducted to compare the results of the data interpretation with and without the use of the algorithm. The DoS, ER, and HCP condition maps for the BEAST deck are shown in Figure 20.

The delamination map from IE is shown in Figure 21. All delamination found was shallow, with a depth of less than approximately 3 cm (1.25 inches). The delamination occurred at the top reinforcement level. The concrete cover thickness was obtained from the GPR survey, as shown in the same figure.

To use the algorithm, the values of the five parameters were input into the prediction table of the algorithm for each point on the BEAST deck test grid. The collected data for the five parameters was organized in an Excel sheet for 1127 points on the BEAST grid. A new set of plots were generated based on the corrected values of the algorithm. Figure 22 shows ER plots before and after using the algorithm for ER. There are significant differences in the values of ER measurements between Figure 22a (before applying the algorithm) and Figure 22b (after applying the algorithm). The measurements changed from around 12 kOhm·cm to approximately 25 kOhm·cm, with the lowest resistivity after using the algorithm being located in the middle of the slab. These changes are primarily due to the influence of the DoS parameter and the effects of delamination. The values seen in Figure 20 of the DoS have a significant impact on the ER values shown in Figure 22b. The algorithm appears to attempt to counteract the effects of DoS by raising the resistivity in areas with high DoS and lowering it in areas with low DoS. Additionally, the IE condition assessment map in Figure 21 shows the location of delamination, which is reflected in the corrected ER values in Figure 22b. The algorithm incorporates the impact of delamination on ER measurement values by decreasing the resistivity value for the delaminated area, which is mainly located in the middle section of the deck.

The way we currently assess corrosion in bridge decks has some limitations. We only collect data at specific times, missing changes that happen in between, and we mainly focus on the surface, not what’s happening inside the concrete. To make this better, future enhancements could incorporate real-time monitoring systems that continuously collect data on factors like moisture levels, temperature fluctuations, and chloride ingress, and the development of algorithms that incorporate a range of material characteristics and environmental conditions.

## 4. Conclusions

This research has illuminated the critical role of a proper interpretation of electrical resistivity (ER) measurements in assessing the vulnerability of reinforced concrete structures to corrosion-induced deterioration. The substantial impact of various influencing factors on the interpretation of ER measurement results was demonstrated, including the degree of saturation, rebar corrosion, and the depth and moisture condition of delamination.

It was also demonstrated that machine-learning techniques are very effective in the development of comprehensive ER measurement correction tools. Such tools are designed to account for the intricate interplay of these influencing parameters, offering a promising avenue for improving the interpretation of ER measurements and the development of a deeper understanding of the relationships between these influencing factors.

The study emphasizes the importance of the meticulous consideration of these various factors in the assessment of reinforced concrete structures. Furthermore, it underscores the tangible benefits that stem from adopting a multi-NDE technology approach in enhancing the reliability and consistency in the condition assessment of bridge decks for better management.

## Figures and Tables

**Figure 1 sensors-23-08052-f001:**
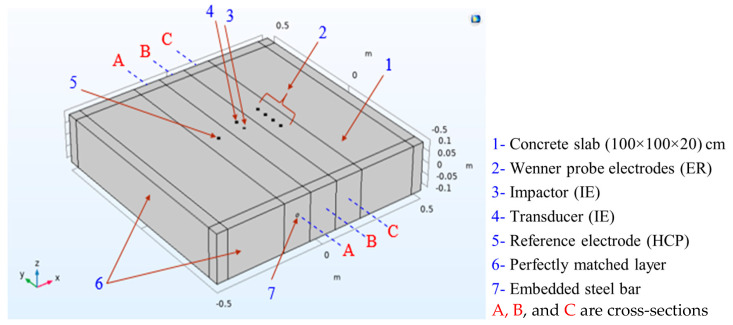
Model components.

**Figure 2 sensors-23-08052-f002:**
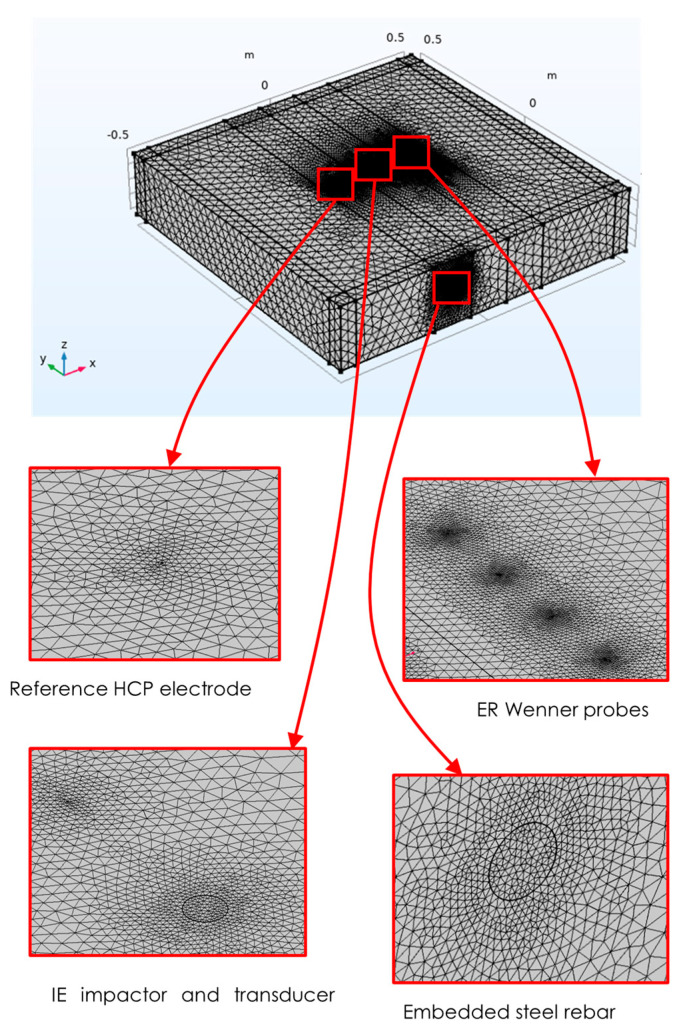
Model meshing.

**Figure 3 sensors-23-08052-f003:**
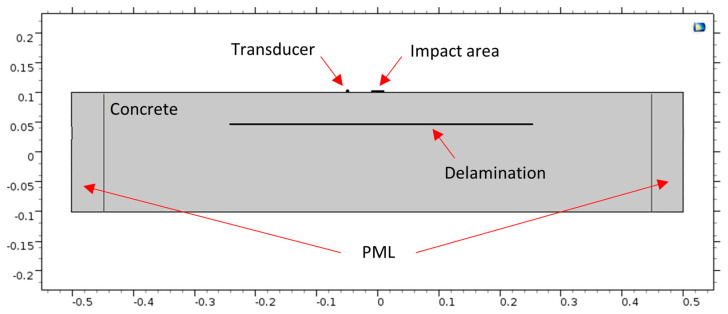
B-B cross section for defective concrete.

**Figure 4 sensors-23-08052-f004:**
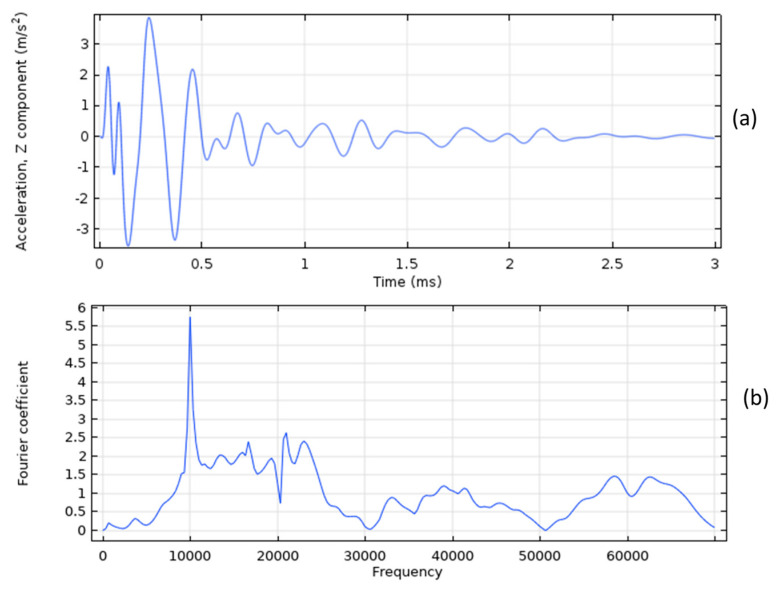
The sound concrete model: (**a**) The acceleration time history, and (**b**) frequency spectrum.

**Figure 5 sensors-23-08052-f005:**
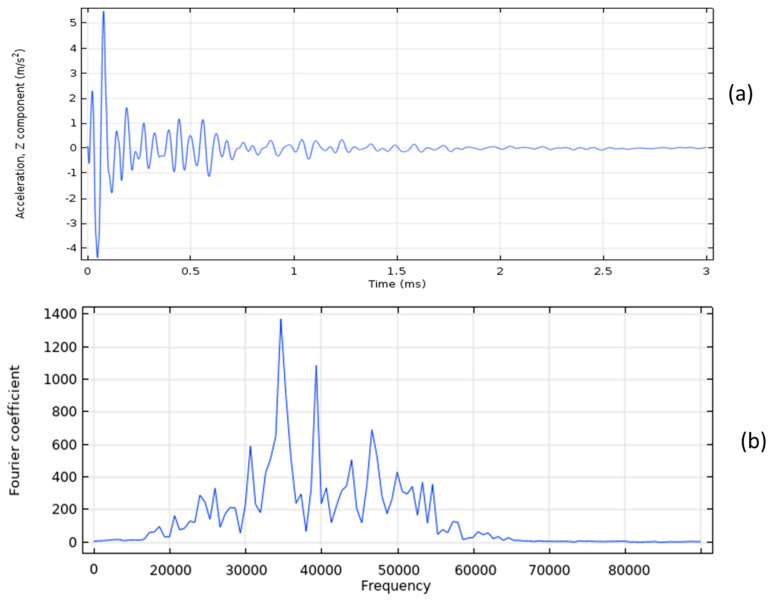
The defective concrete model: (**a**) The acceleration time history, and (**b**) frequency spectrum.

**Figure 6 sensors-23-08052-f006:**
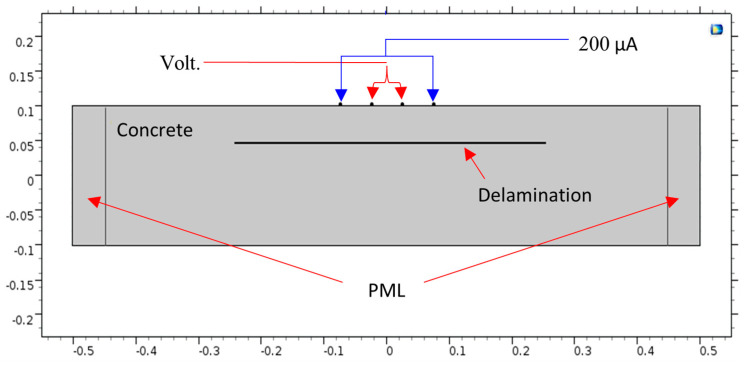
C-C cross-section.

**Figure 7 sensors-23-08052-f007:**
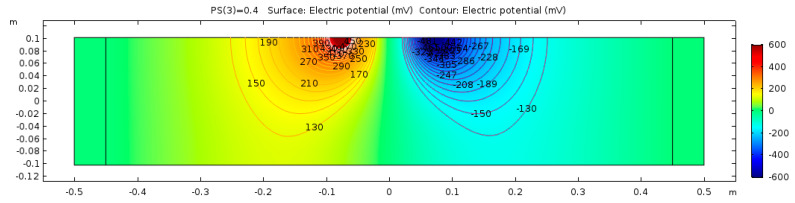
Sound concrete potential distribution from ER simulation.

**Figure 8 sensors-23-08052-f008:**
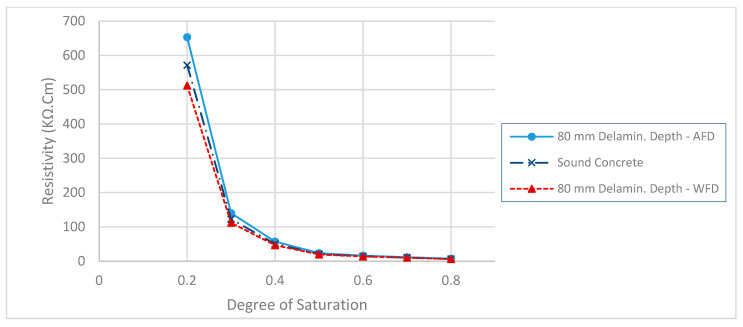
ER for the models with WFD, AFD, and sound concrete.

**Figure 9 sensors-23-08052-f009:**
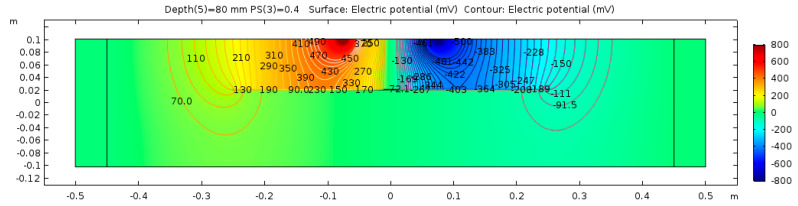
Defective concrete potential distribution from ER simulation—AFD.

**Figure 10 sensors-23-08052-f010:**
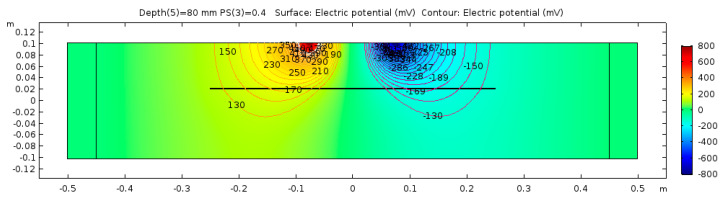
Defective concrete potential distribution from ER simulation—WFD.

**Figure 11 sensors-23-08052-f011:**
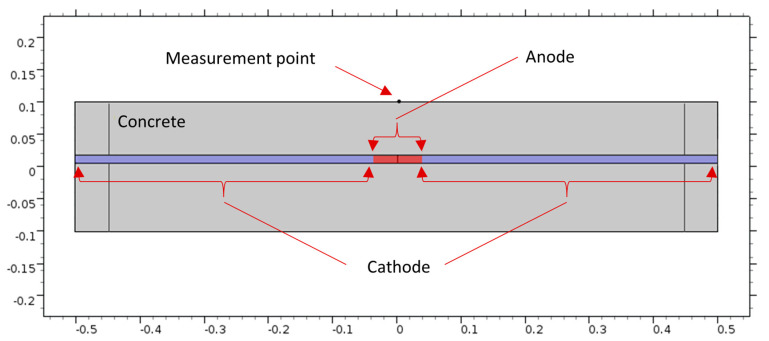
A-A cross-section.

**Figure 12 sensors-23-08052-f012:**
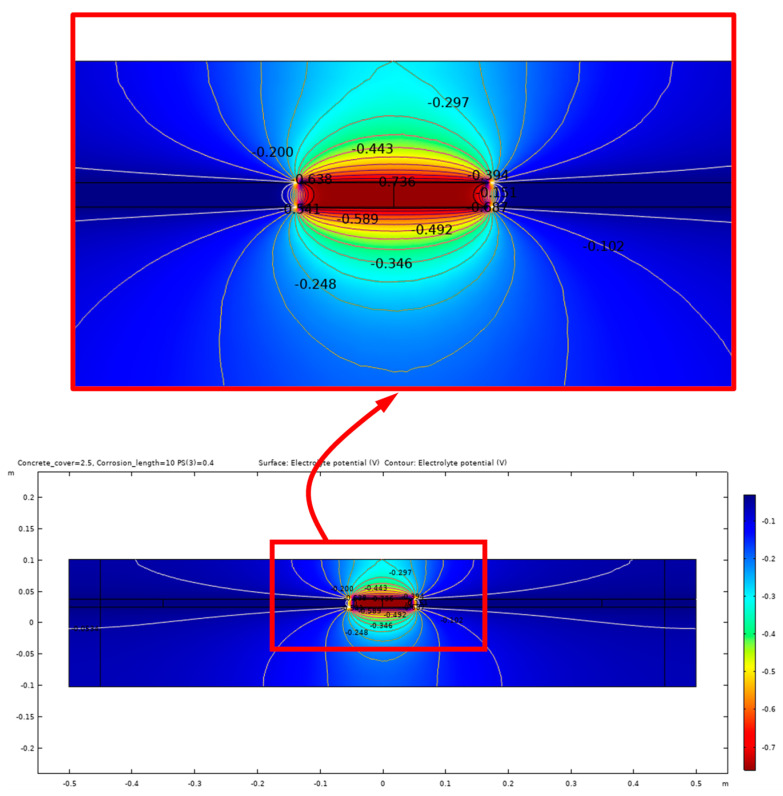
Potential distribution for 10 cm corrosion length, no delamination.

**Figure 13 sensors-23-08052-f013:**
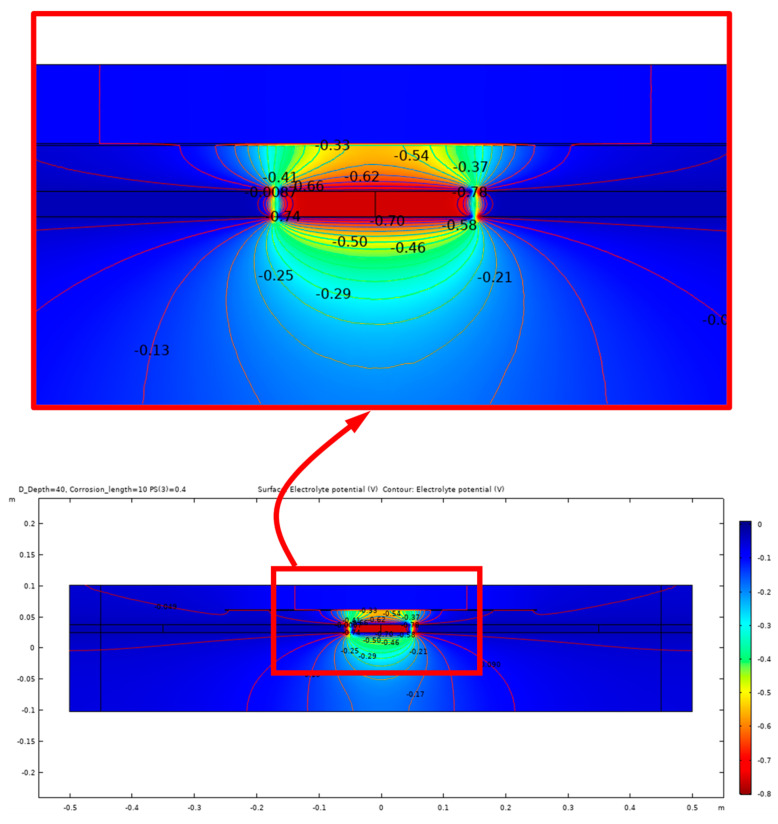
Potential distribution for 10 cm corrosion length, AFD.

**Figure 14 sensors-23-08052-f014:**
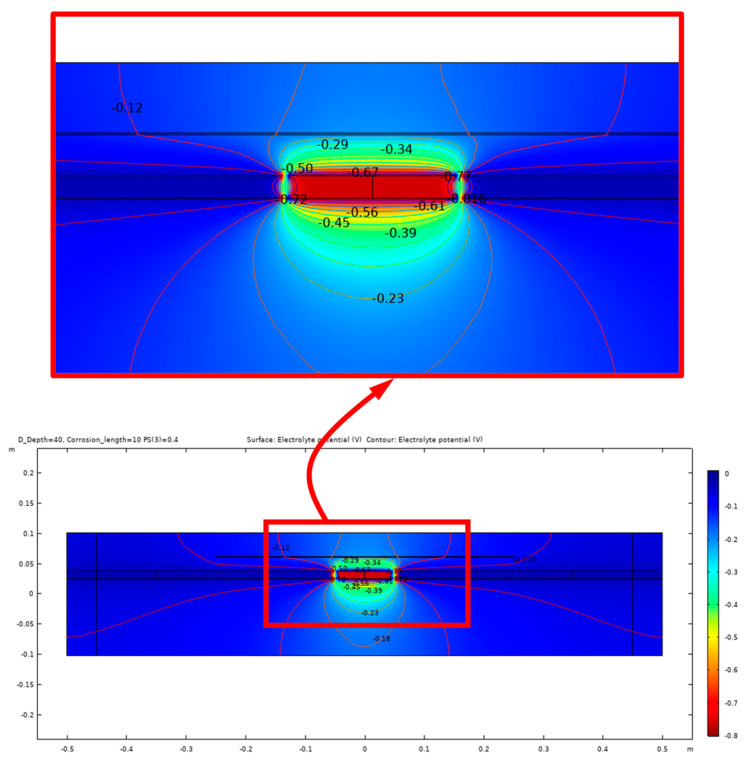
Potential distribution for 10 cm corrosion length, WFD.

**Figure 15 sensors-23-08052-f015:**
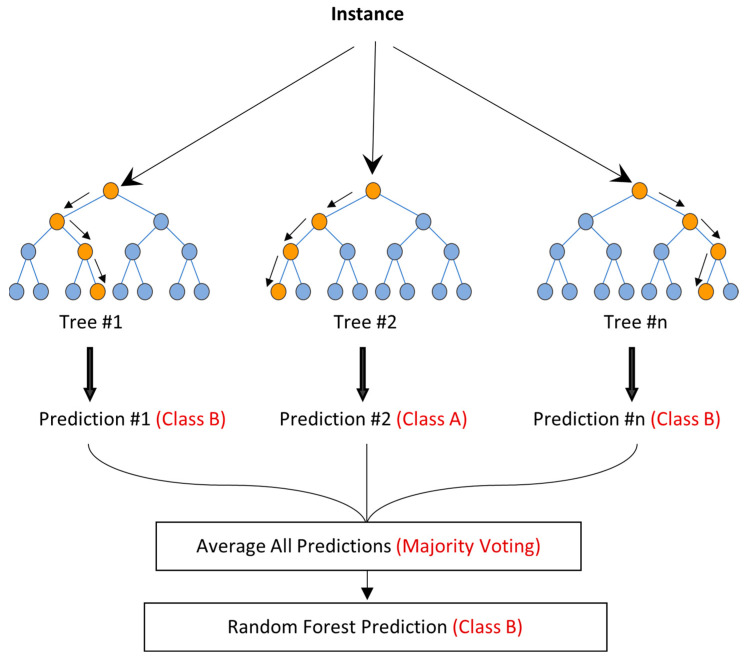
Scheme of random forest regression (black), and classification (red).

**Figure 16 sensors-23-08052-f016:**
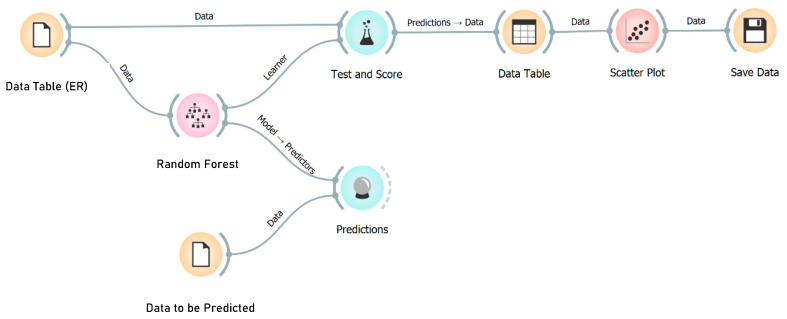
Workflow diagram for the algorithm.

**Figure 17 sensors-23-08052-f017:**
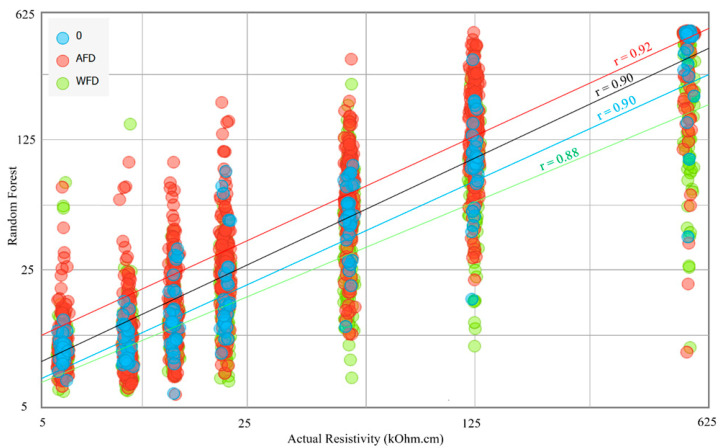
Scatter plot for the Random Forest predictions against the Actual Resistivity.

**Figure 18 sensors-23-08052-f018:**
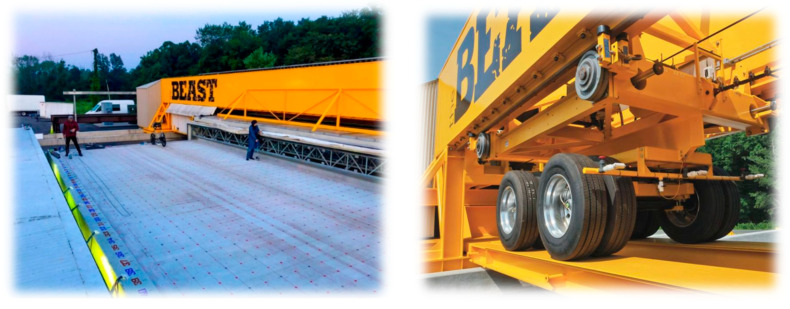
The bridge deck in the BEAST facility (**left**) and the dual-axle carriage (**right**).

**Figure 19 sensors-23-08052-f019:**
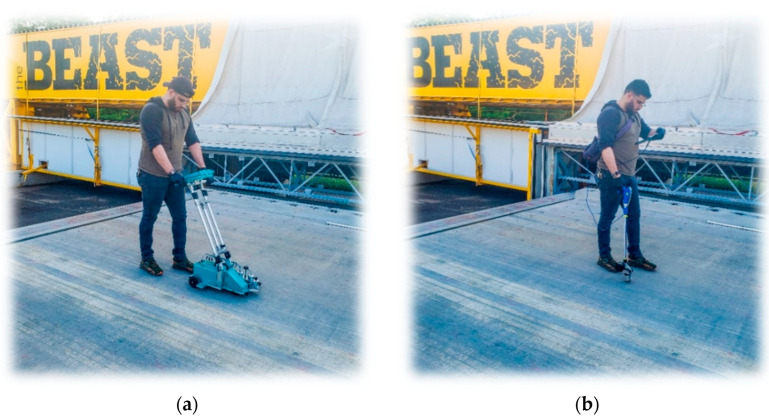
NDE data collection on the BEAST deck: (**a**) MOIST-Scan, (**b**) Impact echo, (**c**) Half-cell potential, and (**d**) Electrical resistivity.

**Figure 20 sensors-23-08052-f020:**
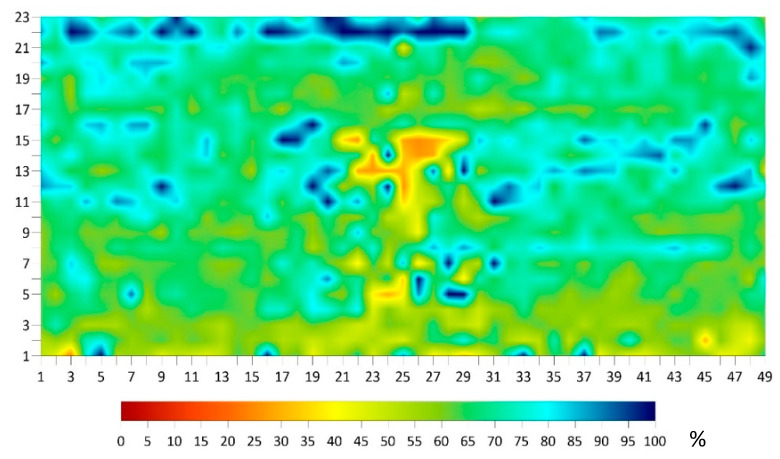
Degree of saturation (**top**), electrical resistivity (**middle**), and half-cell potential (**bottom**) maps from the BEAST slab survey.

**Figure 21 sensors-23-08052-f021:**
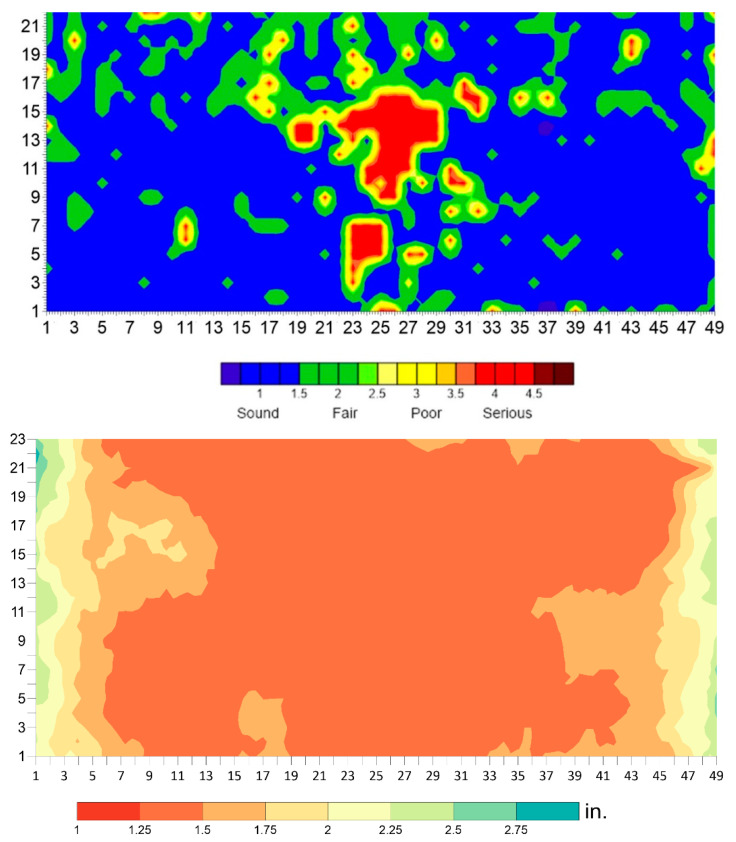
Delamination (**top**) and concrete cover thickness map (**bottom**) of the BEAST deck.

**Figure 22 sensors-23-08052-f022:**
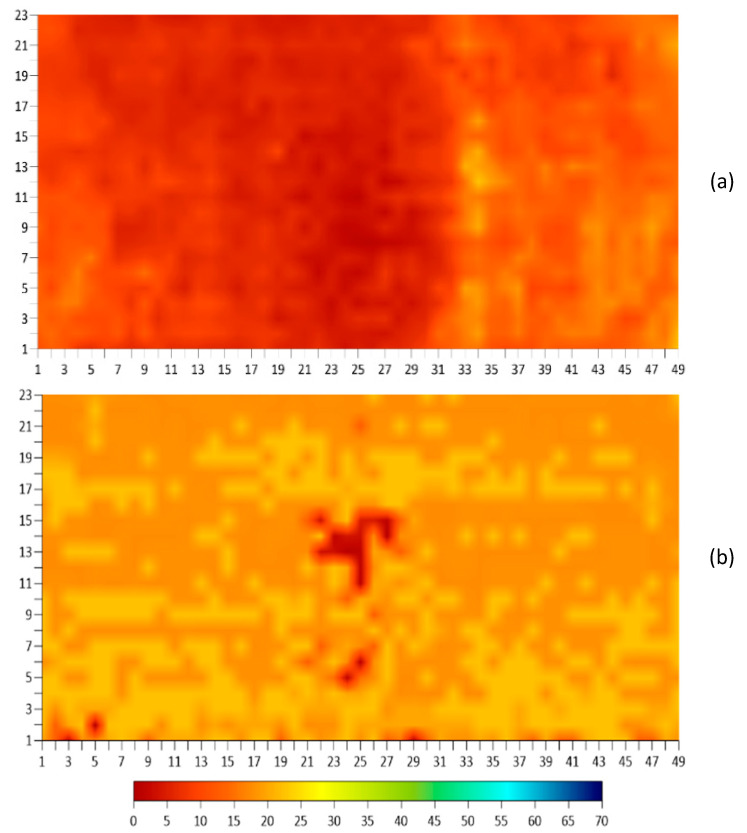
Electrical resistivity results comparison. (**a**) Before and (**b**) after application of the regression algorithm.

**Table 1 sensors-23-08052-t001:** Material properties.

Materials	Properties	
	Electrical Conductivity (S/m)	Relative Permittivity
Concrete	0.002	4.5
Water *	0.5	88.1
Air *	3 × 10^−15^	1

* Used to fill the delamination.

**Table 2 sensors-23-08052-t002:** Concrete Properties.

Property	Value
Density	144.75 lb/ft^3^
Compressive strength	5060 psi
Modulus of Elasticity	3400 ksi
Splitting tensile strength	355 psi
Modulus of rapture	695 psi

## Data Availability

Not applicable.

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
