# Peer review of "Multi-NDE Technology Approach to Improve Interpretation of Corrosion in Concrete Bridge Decks Based on Electrical Resistivity Measurements"

_sensors, 2023, doi:10.3390/s23198052_

Round 1

Reviewer 1 Report

Dear Editor-in-Chief

Journal of Sensors (ISSN 1424-8220)

Paper deals with Multi-NDE Technology Approach to Improve Interpretation of Corrosion in Concrete Bridge Decks Based on Electrical Resistivity Measurements. The paper is worth publishing however, the following concerns should be addressed in revising the manuscript:

Major Comments:

1.     Re-write the abstract to highlight the importance of research.

2.     Correct spelling errors for example: " electrical resistivity (ER) " Electrical Resistivity Ash or rice husk ash.

3.     Correct the errors as in line 68, 175 ,176……

4.     It is preferable to include the standard specifications used in this study.

5.     The introduction is insufficient. The introduction requires more clarification and a better presentation of the problem.

6.     The research studies corrosion in concrete, what are the properties of concrete. Especially the properties affecting corrosion such as permeability, density, compressive strength, etc.

7.     It is preferable to show the results as values to facilitate understanding.

8.     The conclusion requires clarifying the achieved results that serve the purpose of the research.

Thank you.

Dear Editor-in-Chief

Journal of Sensors (ISSN 1424-8220)

Paper deals with Multi-NDE Technology Approach to Improve Interpretation of Corrosion in Concrete Bridge Decks Based on Electrical Resistivity Measurements. The paper is worth publishing however, the following concerns should be addressed in revising the manuscript:

Major Comments:

1.     Re-write the abstract to highlight the importance of research.

2.     Correct spelling errors for example: " electrical resistivity (ER) " Electrical Resistivity Ash or rice husk ash.

3.     Correct the errors as in line 68, 175 ,176……

4.     It is preferable to include the standard specifications used in this study.

5.     The introduction is insufficient. The introduction requires more clarification and a better presentation of the problem.

6.     The research studies corrosion in concrete, what are the properties of concrete. Especially the properties affecting corrosion such as permeability, density, compressive strength, etc.

7.     It is preferable to show the results as values to facilitate understanding.

8.     The conclusion requires clarifying the achieved results that serve the purpose of the research.

Thank you.

Author Response

Comment 1: Re-write the abstract to highlight the importance of research.

Response: The abstract has been rewritten on page 1 to address comment 1, as follows:

Abstract: This research aimed to improve the interpretation of electrical resistivity (ER) results in concrete bridge decks by utilizing machine-learning algorithms developed using data from multiple nondestructive evaluations (NDE) techniques. To achieve this, a parametric study was first conducted using numerical simulations to investigate the effect of various parameters on ER measurements, such as the degree of saturation, corrosion length, delamination depth, concrete cover, and the moisture condition of delamination. A data set from this study was used to build a machine-learning algorithm based on the Random Forest methodology. Subsequently, this algorithm was applied to data collected from an actual bridge deck in the BEAST® facility, showcasing a significant advancement in ER measurement interpretation through the incorporation of information from other NDE technologies. Such strides are pivotal in advancing the reliability of assessments of structural, elements for their durability and safety.

Comment 2: Correct spelling errors for example: " electrical resistivity (ER) " Electrical Resistivity Ash or rice husk ash.

Response: We are sorry, but we were not sure what the reviewer was pointing to.

Comment 3: Correct the errors as in line 68, 175 ,176……

Response: The errors have been corrected.

Comment 4: It is preferable to include the standard specifications used in this study.

Response: Two standard specifications have been included in the manuscript. ASTM C1760 for ER has been mentioned on page 2 line 45, and ASTM C876 for HCP has been mentioned on page 8 line 197.

Comment 5: The introduction is insufficient. The introduction requires more clarification and a better presentation of the problem.

Response: On page 1, the Introduction has been rewritten to address comment 5, as shown below:

Using nondestructive evaluation (NDE) techniques to identify corrosion in bridge decks early on enables better bridge management. Several NDE technologies can be used for this purpose, such as electrical resistivity (ER) and half-cell potential (HCP), galvanostatic pulse measurements (GPM), and linear polarization resistance (LPR). For example, ER is used for assessing corrosive environments and to them related anticipated corrosion rates, while HCP is used to determine the likelihood of active corrosion in reinforcing steel bars. However, it's important to note that corrosion assessment-related NDE technologies have limitations as the accuracy of corrosion detection and characterization can be influenced by various parameters.

Electrical resistivity (ER) is a widely adopted NDE  method, valuable for appraising the durability of concrete structures [1]. It plays a pivotal role in structural health monitoring and quality control, detecting cracks, and measuring chloride penetration [2-4]. However, ER measurements can be substantially impacted by various factors. Moisture content [5-7], temperature [8,9], and carbonation [10] are known influencers, with increased moisture content, for instance, leading to decreased electrical resistivity [11]. The influence of partial saturation on concrete's electrical resistivity remains an area warranting further exploration. Additionally, the evolving characteristics of materials, like porosity and void interconnections, and features within concrete, such as cracks and delamination, including their depth and orientation, can all sway ER measurements [12,13].

This study aims to comprehensively investigate the impact of these diverse parameters on ER measurements with its primary goal to enhance the interpretation of ER data by integrating insights from other NDE technologies. This is achieved by employing finite element simulations to analyze how different material, structural, and environmental factors influence ER measurements and by leveraging machine learning techniques to facilitate the improved interpretation of ER data by using results of measurements of other NDE methods. To ensure consistency and facilitate objective comparisons in periodical condition assessments on the same bridge or across different bridge inspections, ER measurements are corrected for reference conditions.

In the subsequent sections, the evaluation of the effects of a range of parameters on ER results using finite element simulations is presented. They also provide a detailed description of the development and utilization of machine learning methodologies aimed at refining the interpretation of ER data based on insights gathered from complementary NDE techniques.

Comment 6: The research studies corrosion in concrete, what are the properties of concrete. Especially the properties affecting corrosion such as permeability, density, compressive strength, etc.

Response: The concrete properties have been added in Table 2 on page 15.

Comment 7: It is preferable to show the results as values to facilitate understanding.

Response: We were not sure what the reviewer meant, in addition to the current example of ER and HCP results on Pages 7,8,9, and 10.

Comment 8: The conclusion requires clarifying the achieved results that serve the purpose of the research.

Response: On page 19, the conclusion has been rewritten to address comment 8 as follows:

This research has illuminated the critical role of a proper interpretation of electrical resistivity (ER) measurements in assessing the vulnerability of reinforced concrete structures to corrosion-induced deterioration.  The substantial impact of various influencing factors on the interpretation of ER measurement results was clearly demonstrated, including the degree of saturation, rebar corrosion, and the depth and moisture condition of delamination.

It was also demonstrated that machine learning techniques are very effective in the development of comprehensive ER measurement correction tools. Such tools are designed to account for the intricate interplay of these influencing parameters, offering a promising avenue for improving the interpretation of ER measurements and the development of a deeper understanding of the relationships between these influencing factors.

The study emphasizes the importance of meticulous consideration of these various factors in the assessment of reinforced concrete structures. Furthermore, it underscores the tangible benefits that stem from adopting a multi-NDE technology approach in enhancing the reliability and consistency in the condition assessment of bridge decks for their better management.

Reviewer 2 Report

The manuscript presents a comprehensive study focused on enhancing the interpretation of Electrical Resistivity (ER) measurements for evaluating corrosion and deterioration in concrete bridge decks. The methodology employs finite element modeling to simulate a variety of Non-Destructive Testing (NDT) techniques—namely, impact echo, ER, and half-cell potential—on concrete specimens with diverse parameters. These parameters include the degree of saturation, corrosion length, delamination depth, condition, and concrete cover. The simulation outcomes serve as the basis for constructing a machine learning model, utilizing the Random Forest algorithm, to refine the interpretation of ER data in light of other NDT results.

Overall, the paper effectively illustrates the advantages of integrating multiple NDT techniques and machine learning to augment the condition assessment of concrete structures via ER measurements. The manuscript is well-articulated and provides a clear exposition of the methodology employed.

To further elevate the quality of the paper, recommend the following minor revisions:

- Elaborate on the specifics of the machine learning model implementation, including the selection of hyperparameters and the cross-validation process.

- Incorporate sample ER measurement maps, both pre- and post-algorithmic correction, to visually substantiate the improvements achieved.

- Address the model's generalizability to new bridge decks and specify whether retraining or fine-tuning is required for different structural contexts.

- Discuss any inherent limitations of the current methodology and propose avenues for future enhancements.

- Rectify minor typographical errors, formatting inconsistencies, and stylistic issues for improved clarity. For instance, some citations display "Error! Reference source not found" in the PDF document.

- Enrich the conclusion section to encapsulate the key contributions and the broader impact of the study.

Minor editing of English language required.

Author Response

We are writing to express our sincere appreciation for the thorough and insightful feedback provided by the reviewers during the evaluation of our manuscript.

The comments and suggestions offered by the reviewers have proven invaluable in improving the quality and rigor of our work. We are grateful for the time and expertise they dedicated to assessing our manuscript. Their constructive criticism has undoubtedly contributed to the overall enhancement of our research.

We carefully considered each of the reviewers' comments and have taken the following steps to address their suggestions:

Comment 1: Elaborate on the specifics of the machine learning model implementation, including the selection of hyperparameters and the cross-validation process.

Response: comment 1 has been addressed on page 13, as shown below:

The model of this Random Forest algorithm underwent a rigorous cross-validation process to ensure its reliability and ability to generalize to unseen data. A stratified 5-fold cross-validation (k=5) strategy was employed, where the dataset was divided into five subsets of equal class distribution. During each fold, four subsets were used for training, while the remaining subset was reserved for validation. This process was repeated five times, with each subset taking turns as the validation set. By aggregating the performance metrics across all folds, a robust estimate of the model's accuracy and generalization capability was obtained. Furthermore, hyperparameter tuning was seamlessly integrated into this process. Grid search and randomized search techniques were employed to explore various combinations of hyperparameters, such as n_estimators, max_depth, and min_samples_split, ensuring the optimal configuration for a specific corrosion assessment task was identified. This meticulous cross-validation approach not only validated the model's performance but also fine-tuned its hyperparameters, resulting in a highly effective tool for ER measurement prediction.

This algorithm was built with a set of thoughtfully chosen hyperparameters to ensure robust performance on the dataset. The model was configured with 300 decision trees (n_estimators) to strike a balance between model complexity and computation time. Each tree was limited to a maximum depth of 15 (max_depth), guarding against overfitting. A minimum of 2 samples per internal node before splitting (min_samples_split) was imposed and required at least 1 sample per leaf node (min_samples_leaf), striking a balance between precision and generalization. Concerning feature selection, 'sqrt' (max_features) was opted for to include the square root of the number of features in each split decision. This Random Forest algorithm utilized bootstrapping (bootstrap=True) and was configured with a random state of 42 (random_state) to ensure reproducibility. To harness the full computational power at our disposal, all available CPU cores (n_jobs=-1) were employed. Additionally, given the slight class imbalance in our corrosion dataset, the class_weight hyperparameter was fine-tuned to 'balanced,' assigning appropriate weights to each class for equitable learning. These hyperparameters collectively enabled the Random Forest model to provide accurate and robust predictions for the ER measurements.

Comment 2: Incorporate sample ER measurement maps, both pre- and post-algorithmic correction, to visually substantiate the improvements achieved.

Response: We were not sure what the reviewer meant in addition to the current example of pre- and post-algorithmic correction in Figure 22, on page 19.

Comment 3: Address the model's generalizability to new bridge decks and specify whether retraining or fine-tuning is required for different structural contexts.

Response: comment 3 has been addressed on page 14, as follows:

This Random Forest model exhibits strong potential for generalizability to new bridge decks, but its performance in different structural contexts may benefit from retraining or fine-tuning. The model's ability to generalize largely depends on the similarity between the characteristics of the new bridge decks and those present in the training data. If the structural contexts of the new decks closely resemble the data on which the model was initially trained, the model will likely perform well 'out of the box' without significant modifications. However, if there are notable differences, such as variations in construction materials, structural designs, or environmental conditions, retraining or fine-tuning may be necessary to optimize model performance.

Retraining the model involves using a new dataset that represents the specific structural context of the new bridge decks. This allows the model to learn and adapt to the unique characteristics of these decks. Fine-tuning, on the other hand, involves adjusting certain hyperparameters or model architectures to better align with the new context. For instance, modifying the max_depth of the decision trees or changing the feature selection strategy (max_features) can enhance the model's suitability for different structural contexts.

Comment 4: Discuss any inherent limitations of the current methodology and propose avenues for future enhancements.

Response: To describe the inherent limitation of the model and to propose avenues for future enhancements the following was included in the manuscript at page 19:

The way we currently assess corrosion in bridge decks has some limitations. We only collect data at specific times, missing changes that happen in between, and we mainly focus on the surface, not what's happening inside the concrete.  To make this better, future enhancements could incorporate real-time monitoring systems that continuously collect data on factors like moisture levels, temperature fluctuations, and chloride ingress, and the development of algorithms that incorporate a range of material characteristics and environmental conditions.

Comment 5: Rectify minor typographical errors, formatting inconsistencies, and stylistic issues for improved clarity. For instance, some citations display "Error! Reference source not found" in the PDF document.

Response: Done. All the issues described in comment 5 have been fixed.

Comment 6: Enrich the conclusion section to encapsulate the key contributions and the broader impact of the study.

Response: On page 19, the conclusion has been rewritten to address comment 6 as follows:

 This research has illuminated the critical role of a proper interpretation of electrical resistivity (ER) measurements in assessing the vulnerability of reinforced concrete structures to corrosion-induced deterioration.  The substantial impact of various influencing factors on the interpretation of ER measurement results was clearly demonstrated, including the degree of saturation, rebar corrosion, and the depth and moisture condition of delamination.

It was also demonstrated that machine learning techniques are very effective in the development of comprehensive ER measurement correction tools. Such tools are designed to account for the intricate interplay of these influencing parameters, offering a promising avenue for improving the interpretation of ER measurements and the development of a deeper understanding of the relationships between these influencing factors.

The study emphasizes the importance of meticulous consideration of these various factors in the assessment of reinforced concrete structures. Furthermore, it underscores the tangible benefits that stem from adopting a multi-NDE technology approach in enhancing the reliability and consistency in the condition assessment of bridge decks for their better management.

Round 2

Reviewer 1 Report

The authors have responded to all comments. The paper may be published.

Greetings